# Light-induced Kondo-like exciton-spin interaction in neodymium(II) doped hybrid perovskite

Xudong Xiao [1,5], Kyaw Zin Latt [2,5], Jue Gong [1], Taewoo Kim[3], Justin G. Connell [3], Yuzi Liu [2], H. Christopher Fry [2], John E. Pearson[2], Owen S. Wostoupal[1], Mengyuan Li[1], Calvin Soldan[1], Zhenzhen Yang[4], Richard D. Schaller [2], Benjamin T. Diroll [2] ✉, Saw Wai Hla [2] ✉ & Tao Xu [1] ✉

Tuning the properties of a pair of entangled electron and hole in a light-induced exciton is a fundamentally intriguing inquiry for quantum science. Here, using semiconducting hybrid perovskite as an exploratory platform, we discover that $Nd^{2+}$-doped $CH_3NH_3PbI_3$ ($MAPbI_3$) perovskite exhibits a Kondo-like exciton-spin interaction under cryogenic and photoexcitation conditions. The feedback to such interaction between excitons in perovskite and the localized spins in $Nd^{2+}$ is observed as notably prolonged carrier lifetimes measured by time-resolved photoluminescence, ~10 times to that of pristine $MAPbI_3$ without $Nd^{2+}$ dopant. From a mechanistic standpoint, such extended charge separation states are the consequence of the trap state enabled by the antiferromagnetic exchange interaction between the light-induced exciton and the localized $4f$ spins of the $Nd^{2+}$ in the proximity. Importantly, this Kondo-like exciton-spin interaction can be modulated by either increasing $Nd^{2+}$ doping concentration that enhances the coupling between the exciton and $Nd^{2+}$ $4f$ spins as evidenced by elongated carrier lifetime, or by using an external magnetic field that can nullify the spin-dependent exchange interaction therein due to the unified orientations of $Nd^{2+}$ spin angular momentum, thereby leading to exciton recombination at the dynamics comparable to pristine $MAPbI_3$.

The Kondo effect originates from the antiferromagnetic (AFM) exchange interaction (EI) between a localized quantum spin impurity and a large surrounding reservoir of delocalized conduction electrons in a metal host that screens the localized spin[1–4]. In most cases, the Kondo effect has primarily been explored as an electrical transport property often in metal alloys, but occasionally in semiconducting quantum dots for their well-defined spin state applicable in spin-based quantum computing[5,6]. In contrast, integrating the Kondo effect in

optically responsive semiconductors is rarely reported but would also be an attractive approach for light-controlled spintronics because these semiconductors allow non-contact optical injection of delocalized electrons in the conduction band (CB) and/or holes in the valence band (VB). As such, Kondo-like interactions between the impurity spin and the light-induced exciton containing a pair of spin-entangled electron-hole can render a potential optic platform for spintronics and spin-based quantum computing. The prudent use of the term "Kondo-

[1]Department of Chemistry and Biochemistry, Northern Illinois University, DeKalb, Illinois, USA. [2]Center for Nanoscale Materials, Argonne National Laboratory, Lemont, Illinois, USA. [3]Materials Science Division, Argonne National Laboratory, Lemont, Illinois, USA. [4]Chemical Sciences and Engineering Division, Argonne National Laboratory, Lemont, Illinois, USA. [5]These authors contributed equally: Xudong Xiao, Kyaw Zin Latt. ✉e-mail: bdiroll@anl.gov; shla@anl.gov; txu@niu.edu

like" herein is to show both the dissimilarity and similarity between a semiconductor-based optic system and a conventional metal-based Kondo system. There are overwhelmingly more free electrons than spin impurities in a conventional metal-based Kondo system, while there can be much fewer free electrons (or only excitons instead of free carriers) than the doped spin impurities in semiconductors. Both systems involve spin-relevant interaction between carriers and localized spins[7].

The most localized spin to serve as the magnetic impurity is found in the $4f$ orbitals of lanthanides on account of their deformation-induced high spin-orbit coupling (SOC) and the least bandwidth compared with $3d$, $4d$, $5d$, or $5f$ orbitals[8]. Meanwhile, to enhance the direct exchange interaction, the energy level exhibited by the strongly localized $4f$ spin must cross over with the bandgap of the semiconductor host[9,10]. The emerging semiconducting hybrid organic-inorganic perovskites (HOIP) render an excellent platform to dope atomically dispersed magnetic impurities in perovskite lattice via facile wet chemistry[11–14]. More importantly, the CB ($6p$ orbitals) and VB ($5p$ orbitals) of semiconducting lead iodide-based HOIP, such as methylammonium lead triiodide (MAPbI$_3$) respectively concur with the $6s$ $-5d$ hybrid orbital and the $4f$ atomic orbital of lanthanides. Thus, doping MAPbI$_3$ with cations carrying a high and defined number of spin-polarized $4f$ electrons is a promising way to introduce Kondo-relevant spin dynamics around the optically responsive bandgap of MAPbI$_3$, enabling light-switchable spintronics in semiconductors. Nonetheless, such a doping approach also requires a deliberate design using crystal field ligand theory to maximize the spin polarization of the cationic lanthanide dopants in the perovskite host. When bonding with weak ligands, particularly I$^-$ in the MAPbI$_3$ in a hexacoordinated geometry as needed in ABX$_3$ perovskite structure, the $4f$ orbitals of lanthanide cations should have small energy splitting between t$_{2u}$ and a$_{2u}$ levels[15], thereby laying a basis for maximum spin number according to Hund's rules. This being said, such a doping process should be achieved by substituting the perovskite band-relevant B-site divalent metal cation (i.e., Pb$^{2+}$) with a divalent lanthanide cation that possesses the greatest possible number of the same electron spins (maximum of 4) in its t$_{2u}$ and a$_{2u}$ levels per crystal field splitting, as well as a comparable size to Pb$^{2+}$ [15]. These considerations narrow down the selection of lanthanide cations to Nd$^{2+}$ ([Xe]$4f^46s^05d^0$) as the targeted B-site substituent dopant in this work.

Herein, we demonstrate the observation of the interplay between exciton recombination and its Kondo-like interaction with the localized $4f$ spin impurity. Explicitly, the photoelectron in an exciton of MAPbI$_3$ (mainly in the $6p$ orbital of the Pb$^{2+}$ [Xe]$4f^{14}5d^{10}6s^26p^{0\rightarrow1}$) partially hybridizes with the $6s^05d^0$ empty orbital of Nd$^{2+}$, coupling in antiferromagnetic configuration with the localized $4f$ spin in Nd$^{2+}$ via partial intra-atomic exchange interaction[16]. Meanwhile, the photohole in the exciton, essentially the unpaired electron in VB of MAPbI$_3$ (mainly in the $5p$ orbital of I$^-$ [Kr]$4d^{10}5s^25p^{6\rightarrow5}$) couples in antiferromagnetic configuration with another neighboring localized $4f$ spin in Nd$^{2+}$ via inter-atomic exchange interaction. As a result of this Kondo-like interaction, the competing exciton-lattice interaction, i.e., the sub-bandgap trap-induced non-radiative recombination via Shockley–Read–Hall (SRH), Auger recombination, or other non-radiative processes that would otherwise occur in pristine MAPbI$_3$ is largely suppressed in Nd-doped MAPbI$_3$[17,18]. Instead, radiative recombination soars up as evidenced in temperature-dependent steady-state photoluminescence (ss-PL) study. Notably, the photoelectrons and photoholes, when respectively coupled with $4f$ spins in antiferromagnetic configuration, recombine at markedly retarded kinetics observed by temperature-dependent time-resolved PL (tr-PL). Further varying the ratio of Nd$^{2+}$ dopant concentration [Nd$^{2+}$] to incident photon flux allows modulating the population of Kondo-like coupling. The interaction between the photoelectrons and the $6s^05d^0$ empty orbital of Nd$^{2+}$ is locally observed by cryogenic scanning tunneling

microscopy (STM). Our observation is an intriguing optical version of the Kondo-like effect between localized spins and photo-induced excitons of bulk semiconductors at carrier density much less than that of localized spins impurity. This discovery starkly contrasts with the traditional Kondo effect in the steady state of metals, in which the density of free electrons near the Fermi surface is much larger than the spin density of the magnetic impurity.

## Results and discussion

Divalent NdI$_2$ has long been a synthetic challenge but is obtainable by the reported method[19,20], and was used for various Nd$^{2+}$ doped MAPbI$_3$ films (Supplementary Fig. 1) in this work. We first performed the structural and valence electronics characterization of the relevant perovskite films. Figure 1a is the X-ray diffraction (XRD) patterns of 2 mol% Nd$^{2+}$-doped MAPbI$_3$ (denoted as 2%Nd:MAPbI$_3$) versus pure MAPbI$_3$ films, showing the crystal structure in MAPbI$_3$ reserved upon Nd$^{2+}$ doping, thus indicating Nd$^{2+}$ truly replaced Pb$^{2+}$. The ratio of Nd$^{2+}$ to Pb$^{2+}$ is confirmed by inductively coupled plasma mass spectrometry (ICP-MS, see Supplementary Table 1). The Nd$^{2+}$-to-Nd$^{2+}$ distance in 2% Nd:MAPbI$_3$ is about 5.9 nm, exceeding the distance applicable for Ruderman–Kittel–Kasuya–Yosida (RKKY) indirect exchange interaction between neighboring magnetic spins mediated by conduction electrons[21,22]. The slightly shifted (110) peak toward a smaller 2θ angle in the 2%Nd:MAPbI$_3$ film is due to the larger ionic radius (1.30 Å) of Nd$^{2+}$ than that of Pb$^{2+}$ (1.19 Å)[23,24], causing lattice expansion (see XRD refinement in Supplementary Fig. 2 and Supplementary Table 2). Further analysis by the Scherrer equation suggests that the crystallinity in both pristine MAPbI$_3$ and 2%Nd:MAPbI$_3$ films are nearly identical (Supplementary Table 3).

Film structure was further investigated by temperature-dependent selected area electron diffraction (SAED). As seen in Fig. 1b, at room temperature (r.t.), both pristine MAPbI$_3$ and 2% Nd:MAPbI$_3$ thin film exhibit nearly identical microscopic images and polycrystalline features in SAED pattern, belonging to the tetragonal phase except that the 2%Nd:MAPbI$_3$ film shows a slightly increased lattice constant, in agreement with the XRD study. As temperature decreases to 20 K, the orthorhombic phase becomes predominant in both samples based on the crystal plane analysis of the SAED patterns. The (202) plane of 2%Nd:MAPbI$_3$ shows a slight shift towards smaller reciprocal space, suggesting an increased lattice constant compared with the same plane in pure MAPbI$_3$. Other SAED patterns at 40 K to 230 K are shown in Supplementary Fig. 3. The SAED study suggests that both pristine MAPbI$_3$ and 2%Nd:MAPbI$_3$ films exhibit a very similar trend in structural phase transition occurring at the same temperature range and there is no additional structural phase overserved in the 2% Nd:MAPbI$_3$ film within the measured temperature range (r.t. to 20 K), see Supplementary Fig. 4. Therefore, XRD and SAED study clearly suggest that both pure and Nd$^{2+}$ doped samples structurally resemble each other at least between r.t and 20 K and the structure is reversible for both films within at least one cooling cycle. The study of the film morphology and homogeneity of the Nd$^{2+}$ doping sample is further confirmed by the scanning electron microscopy (SEM) and its associated energy dispersive X-ray (EDX) mapping of Pb, I, and Nd, showing a uniform distribution of these three elements across the film. (Supplementary Fig. 5).

The oxidation state of Nd$^{2+}$ is further verified by X-ray photoelectron spectroscopy (XPS) as shown in Fig. 1c. It is clear that the Nd-3d binding energy in 2%Nd:MAPbI$_3$ (1000.3 eV for $3d_{3/2}$ and 978.9 eV for $3d_{5/2}$) is lower than that of reference Nd$^{3+}$ in Nd(NO$_3$)$_3$ (1005.4 eV for $3d_{3/2}$ and 982.8 eV for $3d_{5/2}$)[25], but higher than that of the reported Nd metal (999.1 eV for $3d_{3/2}$ and 978.0 eV for $3d_{5/2}$)[26], signifying the oxidation state of Nd$^{2+}$ setting between that of Nd$^{3+}$ and Nd$^0$. Moreover, due to the low electron negativity of iodine, the binding energy of Nd$^{2+}$ in 2%Nd:MAPbI$_3$ leans towards the metal Nd(0), similar to the trend when Pb$^{2+}$ in MAPbI$_3$ compared with Pb$^{2+}$ in PbO[27].

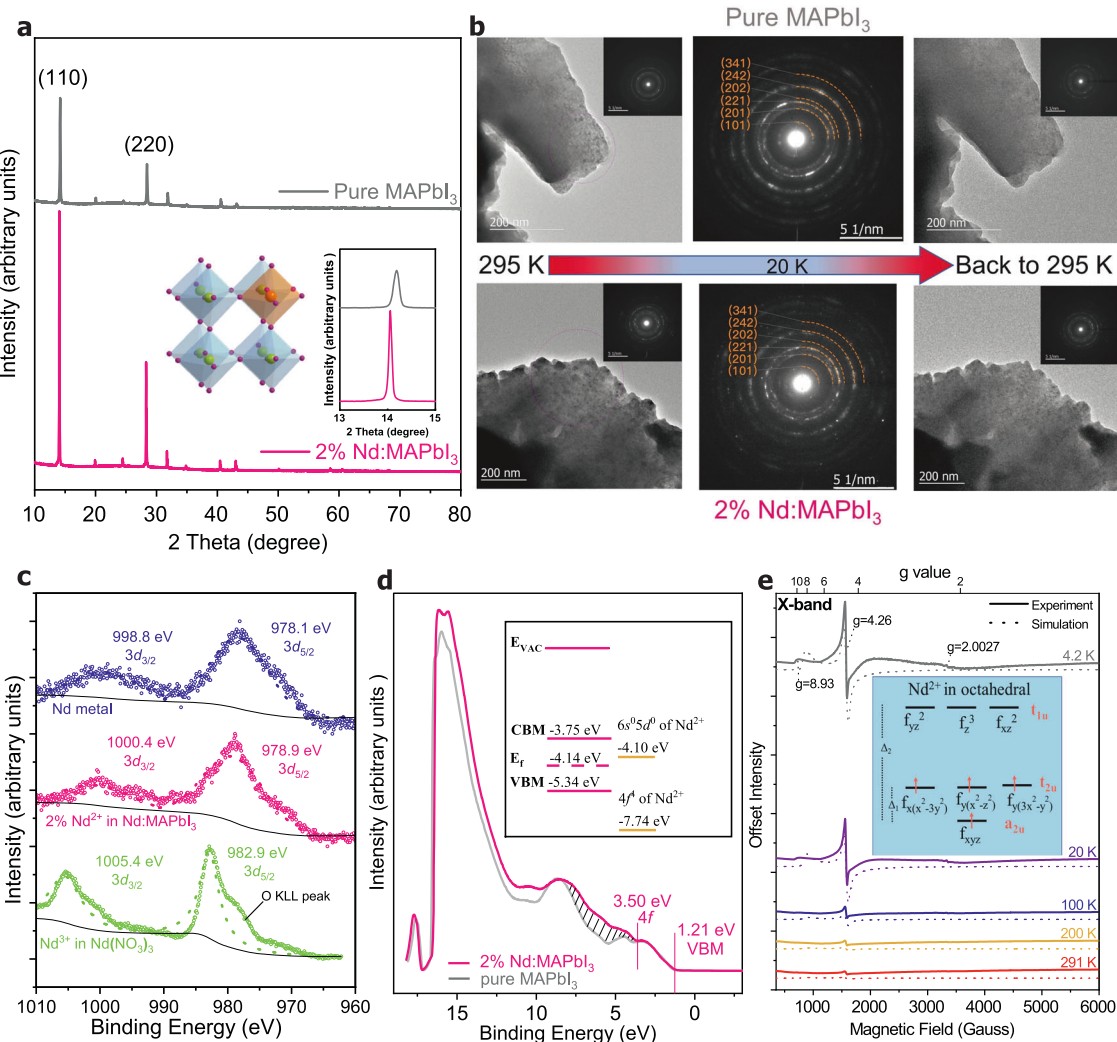

**Fig. 1 | Characterization of pristine MAPbI₃ and 2%Nd:MAPbI₃ films. a** XRD patterns of pristine MAPbI₃ and 2%Nd:MAPbI₃ thin films. The left inset is a schematic illustration of Pb²⁺ (green) substituted by Nd²⁺ (orange) in a perovskite octahedral framework, where I⁻ is in purple and methylammonium (MA⁺) is in blue. The right inset is the magnified comparison for the peak at (110). **b** TEM images and SAED patterns of pristine MAPbI₃ and 2%Nd:MAPbI₃ thin films at r.t. (before cooling), 20 K and warmed back to r.t. (after cooling). **c** XPS comparison of Nd-3*d* for the Nd²⁺ in 2%Nd:MAPbI₃ vs. Nd³⁺ in Nd(NO₃)₃ and Nd metal. Dashed lines are the fitted results. **d** UPS comparison of 2%Nd:MAPbI₃ vs. pristine MAPbI₃. **e** Temperature-dependent CW-EPR spectra of grinded powder of 2%Nd:MAPbI₃. The dotted curves are the simulated results by Easy Spin. The intensities of all curves are offset for a better view. The inset is the proposed high spin ($S = 2$) electron configuration of Nd²⁺ (4$f^4$) in an octahedral crystal field with I⁻ being the weak ligand. $\Delta_1$ and $\Delta_2$ are the crystal field splitting energy between t₂ᵤ and a₂ᵤ energy levels, and between t₁ᵤ and a₂ᵤ energy levels, respectively.

Figure 1d is the valence band structure of pristine MAPbI₃ and 2% Nd:MAPbI₃ films measured by ultraviolet photoelectron spectroscopy (UPS) at r.t. Atop the VB formed mainly by 5*p* band of I⁻ with minor contribution from 6 s band of Pb²⁺. the 4*f* bands from Nd²⁺ dopants (the shaded area) is clearly identified, agreeing with the reported value for Nd-containing compounds[28]. The low binding energy cut-off of this 4*f* band is approximately 2.40 eV below the VB maximum (VBM). Both samples exhibit nearly identical valence band edges in the lower binding energy region, indicating the negligible contribution from Nd²⁺ to the VBM at r.t. Supplementary Fig. 6 entails the analysis of relevant energy diagrams. Since the 6*s5d* band in metallic Nd is about 3.6 eV above the highest cut-off binding energy of the 4*f* band[29], the 6*s5d* of Nd²⁺ is thus estimated to be about −4.1 eV vs vacuum. These relevant energy levels of MAPbI₃ including VBM, CB maximal (CBM), and Fermi energy, and the 4*f* and 6*s5d* orbitals of Nd²⁺ are illustrated in the inset of Fig. 1d. This energy diagram suggests that there is an energy crossover between the bandgap of MAPbI₃ and the highest occupied (4*f*) and the lowest unoccupied atomic orbitals of Nd²⁺, providing an energetically responsive platform for exchange

interaction between the delocalized charge carriers at the band edges of the MAPbI₃ host and the localized 4*f* spin dopants.

Figure 1e is the temperature-dependent X-band continuous-wave electron paramagnetic resonance (CW-EPR) spectra of 2%Nd:MAPbI₃ powder in the dark. In general, the EPR signals are broad and do not exhibit hyperfine structures, indicating that Nd²⁺ is very likely a non-Kramer ion with $S = 1$ if in low spin configuration or $S = 2$ if in high spin configuration. All EPR spectra at different temperatures exhibit similar features with $g_x = g_y = 4.26$, $g_z = 8.93$ due to the zero-field splitting (ZFS) of Nd²⁺. Explicitly, for Nd²⁺ with a 4$f^4$ electron configuration ($S = 2$, $L = 6$, $J = 4$), electron dipole interaction can split the multi-fold degenerated ground state of the electron spin system in the absence of an external magnetic field. The corresponding spin Hamiltonian under ZFS is: $\mathcal{H}_{ZFS} = D[Sz^2 - S(S+1)/3] + E(Sx^2 + Sy^2)$, where $D$ and $E$ are the parameters of ZFS, $D$ describes the axial component of the magnetic dipole-dipole interaction, and E is the transversal component. At 4 K, the ZFS parameters for the simulation (the dotted line is the simulated spectra by MATLAB using the software package EasySpin) are $D = 1.334$ cm⁻¹, $E = 0.045$ cm⁻¹. The weak signal at $g = 8.93$ is associated

with the transition between $m_s = \pm 2$, while the resonance at $g = 4.26$ is due to the transition from $m_s = 0$ to $m_s = 1$[30]. The high-field signal appearing at $g = 2.003$ is very likely due to the lead vacancies with unpaired electron spin (note that the EPR samples are grinded powders in order to fit into the thin EPR test tube), very similar to that of the $Ti^{4+}$ vacancy in $BaTiO_3$ perovskite[31]. This is further confirmed by the CW-EPR test of pristine $MAPbI_3$ powder (Supplementary Fig. 7). The EPR result suggests that there are four spins in the $t_{2u}$ and $a_{2u}$ sublevels, both of which are half-filled, hence, the Jahn-Teller effect can also be excluded. The EPR study, therefore, manifests the high-spin and low-field configuration of the $4f$ electrons in $Nd^{2+}$ as depicted in the inset of Fig. 1e. This configuration is also supported by the fact that crystal field splitting is weak for lanthanides due to the high angular momentum of $4f$ orbital and $I^-$ being one of the weakest ligands. In contrast, orbital angular momentum can often be quenched by ligand fields in $3d$ metals. The temperature-dependent CW-EPR under 405 nm laser irradiation (power = 100 mW) was also conducted to investigate the interaction between the light-induced carrier and the localized spin in $2\%Nd:MAPbI_3$. The comparison between relative EPR signal change between dark and light conditions (Supplementary Fig. 8) exhibits a clear phase change at 100 K, below which more relative percentile loss in EPR signal from dark to light conditions than above it, suggesting that there is another origin accountable for the EPR signal loss besides light-induced heat, most likely due to the AFM coupling between light-induced carries and localized impurity $4f$ spins.

Carrier recombination was investigated by both ss-PL and tr-PL. Figure 2a shows the temperature-dependent PL intensity ratios of $2\%$ Nd:doped $MAPbI_3$ to pristine $MAPbI_3$, and the shift of PL peak wavelength referenced to that at r.t., using a 400 nm pulsed laser, which is deliberately selected to avoid exciting any PL in $Nd^{2+}$ [32]. (Pulse duration = 35 fs, repetition rate = 2000 pulses/s, fluence = 0.88 μJ/cm$^2$, equivalent to intensity = 1760 μW/cm$^2$, or photon flux of 2x10$^{12}$ photons/cm$^2$/pulse or $3.6 \times 10^{15}$ photons/cm$^2$/s). Supplementary Fig. 9 includes all full ss-PL spectra at all measured temperatures. A phase transition from tetragonal (T) to orthorhombic (O) for both pristine $MAPbI_3$ and $2\%Nd:MAPbI_3$ occurs between 100 K and 160 K as indicated by SAED (Supplementary Fig. 3 and 4). Phase transition of pristine $MAPbI_3$ is associated with an emerging high-energy PL emission "shoulder" in agreement with literature[33–35]. In contrast, $2\%Nd:MAPbI_3$ shows a blue-shifted PL peak without the "shoulder" feature (Supplementary Fig. 9). It is notable that after phase transition, the PL peak area of $2\%Nd:MAPbI_3$ rises up from ~ $0.41 \times 10^6$ counts·nm to $1.78 \times 10^6$ counts·nm as temperature decreases from 150 K to 5.7 K, in strong contrast to the change in PL peak area of pristine $MAPbI_3$ from $0.37 \times 10^6$ counts nm to $0.67 \times 10^6$ counts nm in the same temperature range. Such enhanced PL intensities strongly suggest a potential application of $Nd^{2+}$ doping in enhancing the PL intensities of bulk perovskite films, in contrast with the conventionally adopted quantum-dot perovskite films for their large emission efficiencies.

The enhanced PL emission is a clear indication of less non-radiative recombination via carrier-phonon interaction in $2\%$ Nd:$MAPbI_3$ than that in pristine $MAPbI_3$. Within the O phase, pristine $MAPbI_3$ shows distinctive high-energy emission peaks due to donor-acceptor-pair transition, as regulated by the trap states within the material[36]. $2\%Nd:MAPbI_3$ in the O phase, however, still exhibits rapid redshift with decreasing temperature, in stark difference from pristine $MAPbI_3$. Figure 2b is the comparison of temperature-dependent average PL lifetime $<\tau>$ (defined in Eq. 1) for $2\%Nd:MAPbI_3$ and pristine $MAPbI_3$ excited under the same conditions. The $2\%Nd:MAPbI_3$ displays a slightly shortened lifetime compared to pristine $MAPbI_3$ at r.t., indicating the thermal energy (kT) still outpaced the quantum spin-spin exchange interaction between the $4f$ spin in $Nd^{2+}$ and the photo-carriers in $MAPbI_3$. As temperature drops, $2\%Nd:MAPbI_3$ shows a nearly monotonic increment of its $<\tau>$ that exceeds 1 μs at 5.7 K. The original tr-PL and fitting details are collected in Supplementary Fig. 10–14. This

observation is strikingly discernable from the continually reduced $<\tau>$ on pristine $MAPbI_3$ upon lowered temperature that agrees well with an earlier report[33], despite a plateau between 150–200 K as related to the structural T to O phase transition[37], with $<\tau>$ eventually reaching < 10 ns at 5.7 K. The temperature-dependent trends of $<\tau>$ for pristine $MAPbI_3$ and $2\%Nd:MAPbI_3$ films were re-confirmed by tr-PL decays obtained by using a diode laser as an excitation source (Supplementary Fig. 15). Although similar PL decay behaviors caused by magnetic polaron formation can be observed in semiconductors doped with magnetic ions[38], they are still different from our finding as the former generates very short spin relaxation dynamics at the scale of picoseconds, and previous report has indicated a suppressed magnetic polaron in bulk semiconductor systems. On the other hand, as our material paradigm requires optical excitation to generate photocarriers (excitons) to imitate the conduction electrons as occurring in classic metal-based Kondo systems, and arise from the alignment of the localized dopants spins with the exciton spins, they also differ from the intrinsic magneto-optical and magneto-electric properties of diluted magnetic semiconductors. It is noteworthy that other factors could lead to elongated PL carrier lifetimes, such as Coulomb interactions, initial distribution of photocarriers in electronic bands, less defect-assisted relaxation, van der Waals structures, and transition among different quasiparticles[39]. But we can mostly exclude these factors because 1) perovskite materials applied in this study are three-dimensional structures with and without atomically dispersed $Nd^{2+}$ dopant, and therefore do not introduce multilayered heterostructures, external dielectric environment, interlayer strain, or charge carriers; 2) identical photoexcitation and acquisition conditions are utilized, plus the Nd energy levels that mostly affect perovskite intragap and VB regions, both pristine and Nd-doped perovskite films exhibit comparable electronic band structures; 3) both samples show comparable ss-PL intensities and tr-PL decay dynamics initially before temperature reduction, and temperature-dependent TEM/SAED images and the derived diffraction patterns (Supplementary Fig. 4) indicate comparable crystal structures, thus signifying comparable densities of structural defects; 4) at cryogenic temperatures, the form of photocarriers remains excitons in the absence of electrical bias or multilayer heterostructures. On the other hand, to comprehend the relaxation of phonon-assisted photocarriers, we examined the temperature-dependent broadening of emissions and extracted the full width at half-maximum (FWHM) from the photoluminescence spectra. Through the analysis of the temperature-dependent emission broadening and extracting the full width at half-maximum (FWHM) of the PL spectra (Supplementary Fig. 16a), one can see that Nd-doped $MAPbI_3$ film indeed exhibits a smaller change of the temperature-dependent PL linewidth compare with the pristine $MAPbI_3$ film. For hybrid lead halide perovskite materials, charge-carrier-phonon interaction is dominated by Frohlich interaction between charge carriers and LO phonons[34], the equation is expressed as:

$$\Gamma_{LO} = \gamma_{LO} \cdot \frac{1}{\left[ e^{E_{LO}/k_B T} - 1 \right]} \qquad (1)$$

Here in Eq. 1, $\Gamma_{LO}$ is results from LO phonon scattering, $\gamma_{LO}$ is the corresponding charge-carrier-phonon coupling strength, and $E_{LO}$ is the Energy representative of the frequency for the weakly dispersive longitudinal optical (LO) phonon branch. The parameters, $\gamma_{LO}$ and $E_{LO}$ can be derived by fitting the temperature-dependent half-maximum (FWHM) of the PL spectra. Therefore, the longitudinal optical phonon energy ($\Gamma_{LO}$) of Nd-doped $MAPbI_3$ as compared to pristine $MAPbI_3$ at different temperatures (Supplementary Fig. 16b) can be exported, thereby substantiating the suppressed carrier-phonon coupling mentioned in the previous context.

The spin-photocarrier interaction can be further modulated by tuning the relative ratio of $[Nd^{2+}]$ per area (denoted as $[Nd^{2+}]_{area}$,

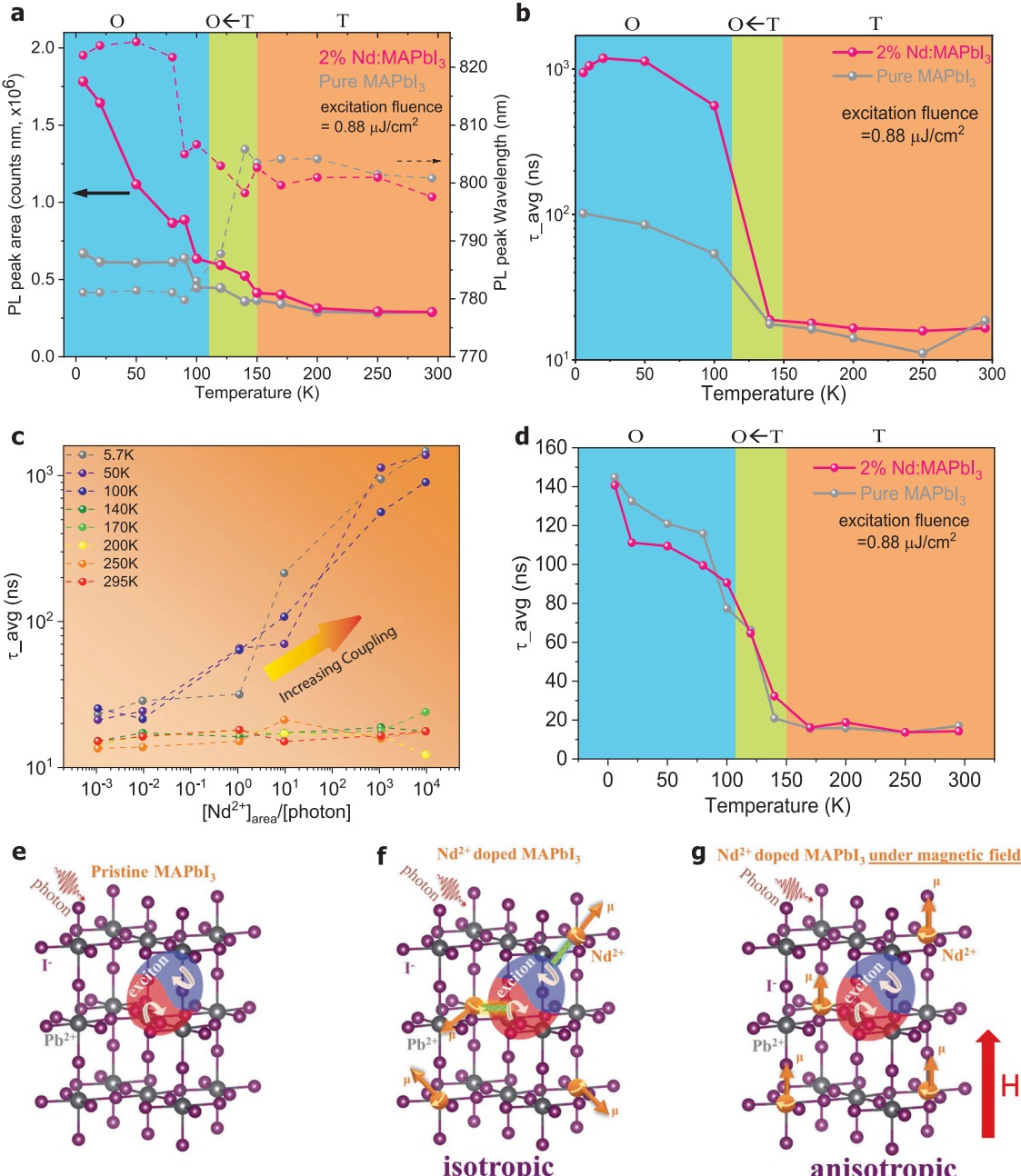

**Fig. 2 | Photoluminescence study of pristine MAPbI₃ and 2%Nd:MAPbI₃ films.**
**a** The temperature-dependent ss-PL peak wavelength, and PL intensity ratio of 2% Nd:MAPbI₃ to pristine MAPbI₃. **b** Temperature-dependent lifetime <τ> of pristine MAPbI₃ vs. 2%Nd:MAPbI₃ extracted from temperature-dependent time-resolved PL decays study. **c** Summary of lifetime <τ> at different temperatures and Nd²⁺-to-photon density ratios. All lines are as a guide to the eye. **d** Temperature-dependent lifetime <τ> of pristine MAPbI₃ vs. 2%Nd:MAPbI₃ under magnetic field (normal to sample surface with magnetic field strength of 1500 Gauss near sample surface)

extracted from temperature-dependent time-resolved PL decays study.
**e** Recombination of photocarriers in pristine MAPbI₃. **f** In the absence of a magnetic field, the spin exchange interactions between the isotropic localized spins of Nd²⁺ cations with the electron and hole in the exciton are allowed to occur respectively, as these interactions do not break the total spin = 0 of the exciton. **g** In the presence of a magnetic field, the localized spins on Nd²⁺ cations are anisotropically polarized, preventing the coupling between Nd²⁺ spin with either electron or hole in the exciton as otherwise the net spin of exciton becomes non-zero.

which is $2.2 \times 10^{15}$ Nd²⁺/cm² for a 2%Nd:MAPbI₃ film thickness of ~300 nm) to that of incident photons. Thus, a low [Nd²⁺] doped 20 ppm-Nd:MAPbI₃ (i.e., molar ratio of Nd:Pb = 20:10⁶) was prepared, equivalent to [Nd²⁺]$_{area}$ = $2.2 \times 10^{12}$ Nd²⁺/cm². A lower excitation intensity of fluence = 0.11 μJ/cm², equivalent to 220 μW/cm² (400 nm, [photon] = $2.2 \times 10^{11}$ photons/cm²/pulse) was also adopted. The combination of two different excitation light intensities and two [Nd²⁺]$_{area}$ allow us to modulate the ratio of [Nd²⁺]$_{area}$/[photon]. Figure 2c shows the temperature impact on the <τ> across the [Nd²⁺]$_{area}$/[photon]

range of $10^{-3} \sim 10^4$. The monotonic increment of <τ> at greater [Nd²⁺]$_{area}$/[photon] ratio below 100 K is clearly evident. At transition between 140 K – 100 K, particularly at a higher [Nd²⁺]$_{area}$/[photon] ratio, can be clearly identified. It is known that in the O phase at a temperature below 120 K, excitons are the dominant light-induced carriers in MAPbI₃[18]. We further studied the PL properties under a fixed permanent magnetic field. First, we methodically studied the magnetization property of 2%Nd:MAPbI₃. The temperature-dependent hysteresis curve of 2%Nd:MAPbI₃ and pristine MAPbI₃

are collected in Supplementary Fig. 17a. It indicates that there is a positive magnetic moment observed, which can be attributed to the alignment of spin moments of Nd ions within the sample. With the decrease in temperature, the magnetization curves displayed nearly S-shaped behaviors, indicating stronger magnetic interactions. In contrast, the pristine MAPbI$_3$ shows diamagnetic behaviors with a small negative response to the external magnetic field. Furthermore, we also studied the magnetic susceptibility as a function of temperature for the 2%Nd:MAPbI$_3$ powder, shown in Supplementary Fig. 17b. By using the Curie-Weiss (CW) law, the effective magnetic moment can be derived as $\mu_{eff} = 4.76\,\mu_B$. This result is similar to the theoretical effective magnetic moment of Nd$^{2+}$ which is $\mu_{eff} = 4.9046\,\mu_B$. The result also indicates the bivalent state of Nd ion in the sample which corresponds to the XPS result. Figure 2d shows that the long PL lifetime in 2%Nd:MAPbI$_3$ sample at a low-temperature range (< 120 K) intriguingly vanishes in a weak magnetic field (0.15 T) normal to sample surface provided by a permanent SmCo magnet (suitable for low-temperature)[40] positioned behind the samples. See the Materials and Methods section for detailed descriptions of the magnetic field effect PL measurement procedures. The temperature-dependent PL lifetime of the 2%Nd:MAPbI$_3$ sample resembles that of the pristine MAPbI$_3$ sample. Both show little difference from that of the pristine MAPbI$_3$ sample in Fig. 2b. To verify the magnetic field effect on the PL spectra of the Nd$^{2+}$ doped sample, we repeated the measurement on the same 2%Nd:MAPbI$_3$ sample. Comparison of the ss-PL spectra of this 2%Nd:MAPbI$_3$ sample with and without magnetic field illustrates the markedly attenuated PL intensities when the magnetic field is present (Supplementary Fig. 18). Supplementary Fig. 19 verified the long PL lifetime in this 2% Nd:MAPbI$_3$ sample in absence of magnetic field at low-temperature range (< 120 K), similar to the 2%Nd:MAPbI$_3$ sample (no magnetic field) in Fig. 2b. However, under magnetic field, the long PL lifetime of this 2%Nd:MAPbI$_3$ sample in the low-temperature range (< 120 K) vanished, and becomes similar to that of the pristine MAPbI$_3$ sample (without magnetic field) as shown in Fig. 2b. On the other hand, we can rule out the possibility that Nd doping leads to lattice disorder and the potential increments in PL intensity and carrier lifetime, despite that structural distortion can in fact, enhance light emission efficiencies in a certain inorganic context such as InGaN multilayer structure[41,42]. The magnetic field should not be able to control the regarded lattice disorder and annihilate/generate structural defects that affect the ss-PL intensity and carrier lifetime as observed. Therefore, the exciton-spin interaction, as valved by the magnetic field and Nd impurity spins should be the responsible mechanism for the abovementioned light-induced observations at cryogenic conditions. As such, the demonstrated elongated PL lifetime as controllable upon Nd$^{2+}$ doping concentration and magnetic field clearly indicates a long but manipulable spin relaxation process that are potentially useable in high-performance spintronics and quantum computing applications, where achieving long coherence time of electron spins is critical for quantum manipulation[43]. The original tr-PL and fitting details are collected in Supplementary Fig. 20 and 21. Figure 2e illustrates the interplay between the recombination of an exciton and its exchange interaction with localized 4$f$ spin, resulting in a metastable pinned exciton and the consequently retarded recombination kinetics. Note that J$_{f-psd}$ is the partial intra-atomic exchange constant, while J$_{f-sd}$ is the inter-atomic exchange constant. It is reported that the radii of an exciton in MAPbI$_3$ are in the range of 3 ~ 5 nm[44], which is large enough to span two neighboring Nd$^{2+}$ dopants (5.9 nm apart aforementioned). Thus, we think it is likely that each exciton can interact with at least two Nd$^{2+}$ in proximity. The schematic, as shown in Fig. 2f, is allowed because of the isotropic magnetic spin moments of localized 4$f$ spins that are randomized in the proximity of an exciton so that the photoelectrons and photoholes in the exciton have a high probability of forming

antiferromagnetic exchange interaction with nearby 4$f$ spins. In contrast, as illustrated in Fig. 2g, when the localized 4$f$ spin magnetic moments are polarized by an external magnetic field, antiferromagnetic exchange interaction between the exciton and its surrounding 4$f$ spins becomes anisotropic with diminished probability or is replaced by weaker ferromagnetic exchange interaction[3]. Thus, more than the relative ratio of [Nd$^{2+}$]$_{area}$/[photon], a magnetic field also acts as a switch to modulate this light-induced Kondo-like coupling.

Next, we investigated the local electronic properties using low-temperature scanning tunneling spectroscopy at 5 K substrate temperatures in an ultrahigh vacuum condition. For the experiments, a photon beam of 400 nm wavelength was illuminated onto the tip-sample junction. The light illumination onto the sample is required for the tip to approach the sample because the recorded tunneling current under the dark condition is significantly lower than that under illumination as shown in supporting information Supplementary Fig. 22. Scanning tunneling microscopy (STM) images of the sample (exemplified in Fig. 3a) do not show large height variations however, detailed surface features are difficult to resolve probably due to the light illumination. Therefore, the point spectroscopic measurements are performed across the surface at different locations without taking local area images. For the measurements, the STM tip is positioned at a fixed height, the bias voltage is ramped from −1 V to +1 V under 400 nm illumination, and the corresponding tunneling current is recorded using a lock-in amplifier. The semiconducting characteristic of the undoped sample areas is clearly observed in the tunneling spectroscopy (I–V curve) data (Fig. 3b) and simultaneously measured differential tunneling conductance (dI/dV-V) spectra (Fig. 3c). Here, the CB and VB gap is measured as ~ 1.5 eV, which agrees well with the expected gap of 1.59 eV. For Nd-doped areas of the 2%Nd:MAPbI$_3$ sample, the dI/dV-V spectroscopy data show a smaller gap-like feature (Fig. 3d, e). Its symmetric nature and the energetic location, ± 0.28 eV, indicate that it is of a different origin. The d$^2$I/dV$^2$ data (Fig. 3f) clearly indicates that the step-like features observed in Fig. 3e are indeed related to the inelastic electron tunneling (IET) process. More complete tunneling spectroscopic data collected at random sample areas reveal both the presence and absence of step-like features in dI/dV and d$^2$I/dV$^2$ curves of Nd-doped MAPbI$_3$ film (Supplementary Figs. 23 and 24). Supplementary Figs. 25 and 26 exhibit another local domain with a more uniform distribution of Nd$^{2+}$ as evidenced by the periodical occurrence of the step-like features. In stark contrast, this step-like feature was never observed from pristine MAPbI$_3$ film that only exhibits the typical bandgap characteristics of MAPbI$_3$ (Supplementary Figs. 27 and 28), suggesting that the IET process only occurs in Nd-doped MAPbI$_3$ film.

It is reasonable to suggest that at the low substrate temperature of 5 K, the photoexcited electrons are being trapped or filled (as evidenced in the optic study as well) in the 6$s^0$5$d^0$ empty orbital of the Nd$^{2+}$ in agreement with the energy gap (~ 0.3 eV) between 6$s^0$5$d^0$ orbital of the Nd$^{2+}$ and the conduction band minimum of MAPbI$_3$ as inferred by UPS results (Figs. 1d & 3g). In general, IET measures the excitation energies of atoms or molecules within the tunnel junction[45–47]. In our case, the 0.28 eV energy is attributed to the excitation of the trapped electrons in this gap (Fig. 3g).

In closing, our work has demonstrated an optical Kondo-like effect in which the density of light-induced delocalized electrons is much outnumbered by the localized spins from magnetic impurity. As such, the spin-entangled electrons and holes have a high probability to respectively couple with the opposite localized impurity spin within their proximity as evidenced by the notably prolonged carrier lifetimes. Interestingly, an external magnetic field nullifies such interaction between the exciton and the localized impurity spin because of the vanishment of opposite local spins needed for the electron and hole, respectively. The formation of the exchange interaction between the delocalized electrons in MAPbI$_3$ and localized spins in Nd$^{2+}$ is

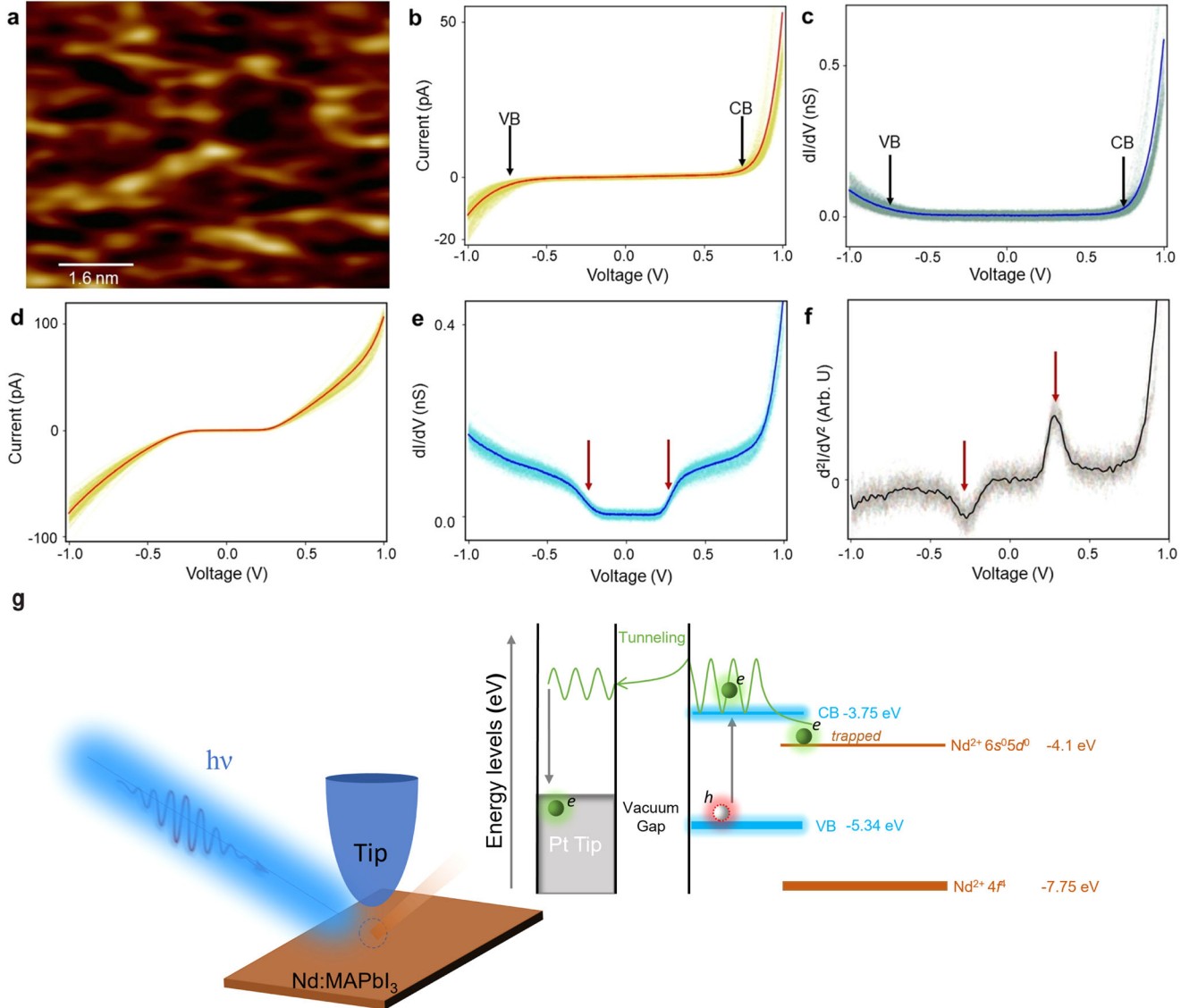

**Fig. 3 | Local electronic structural investigations by cryogenic scanning tunneling microscopy (STM). a** STM image of the sample surface acquired under 400 nm illumination ($V_t = 1$ V, $I_t = 50$ pA). **b** I–V spectroscopy, and (**c**) simultaneously acquired dI/dV-V spectroscopy reveal a semiconducting behavior. **d** I–V spectroscopy, **e** dI/dV-V spectroscopy, and (**f**) d²I/dV²-V plots associated with Nd²⁺ sites. The arrows indicate the change in energy at $\pm 0.28$ V. **g** A demonstration of STM spectroscopy process under 400 nm illumination at the tip-sample junction. The schematic drawing at right illustrates exciton (electron-hole pair) formation upon illumination, and a trapped electron at the Nd²⁺ site (indicated with a dashed circle). The spectroscopy data shown in (**b**–**f**) are generated from 64 separate measurements at different locations across the sample. In each figure, the average curve appears as a single darker-color plot.

antiferromagnetic by nature, a consequence of partial photoelectron trapping in the $6s5d$ orbitals that result in 10 folds extended carrier lifetimes of perovskite films at low temperatures. Importantly, because the concentration of magnetic spins in Nd²⁺ overwhelmingly exceeds the photocarriers in the perovskite host, a case impossible to be achieved in a metal-based classic Kondo process, we are able to control the coupling intensity (embodied by carrier lifetime of perovskite MAPbI₃) through the amount ratio of Nd²⁺ to the incident photons as well as by the external magnetic field. In perspective, our work shows an approach to apply quantum interference on a pair of spin-entangled particles by localized spins within close proximity and inspires the discovery of strongly correlated light-matter interaction with evolutionary states at the localized-itinerating electron crossover regions, where mutual coupling degrees of freedom among charge, spin, orbital and lattice can lead to exotic photon-induced electronic phases and applications in spintronics and many-body entanglement-based quantum computing.

## Methods

### Materials

PbI₂ (99.9985%, Alfa Aesar), methylammonium iodide (MAI, 98%, Sigma Aldrich), neodymium ingot (99.9%, Sigma Aldrich), iodine (99.99%, Sigma Aldrich), dimethyl sulfoxide (DMSO, 99.9%, Sigma Aldrich), N, N-dimethylformamide (DMF, 99.8%, Sigma Aldrich), chlorobenzene (CB, 99.8%, Sigma Aldrich), Nd(NO₃)₃ • 6H₂O (99.9%, Sigma Aldrich), fused silica wafer (University Wafer).

### Synthesis of NdI₂

NdI₂ was prepared in a quartz tubular flask by solid-state reaction. The Neodymium shaving was acquired by using the electrical drill to drill on the neodymium ingot. Iodine powder was acquired by grinding the iodine chunks. Then the neodymium shaving and iodine powder were mixed and poured into the quartz flask. The apparatus was vacuumed for 20 min, and then the valve was closed. The quartz flask was then heated by natural gas flame until the purplish-red glow emitted from

the solid-state reaction. The reaction product, $NdI_2$ was poured into a glass vial after the apparatus was cooled down to room temperature. DMSO as the solvent was added to the glass vial to dissolve $NdI_2$, and an ultrasonicator was used to accelerate the dissolving process. The solid in the mixture was removed by using the PTFE filter twice. The final purified solution is referred to as $NdI_2$/DMSO solution.

## Preparation of perovskite films

The perovskite film was deposited on the fused silica wafer by using the spin coating method. First, the silica wafer substrate was cleaned in deionized water, acetone, and isopropanol, subsequently. Then the substrate was cleaned with a UV-ozone cleaner for 10 min. The perovskite precursor, $MAPbI_3$ solution was prepared from a precursor solution containing MAI (1 M) and $PbI_2$ (1 M) in anhydrous DMF:DMSO in a 4:1 volume ratio. The different volumes of $NdI_2$/DMSO solutions were added into the perovskite precursor to make the different concentrations of lanthanide dopant $MAPbI_3$ perovskite precursors. The perovskite film was spin-coated by a one-step program (30 s at 4000 rpm, ramp: 2000 rpm·s$^{-1}$). 120 μL of chlorobenzene was poured onto the spinning substrate at the 15th second in the program The substrates were then annealed at 100 °C for 10 min in an argon-filled glove box.

Powder sample preparation for EPR and XRD measurement: 2% Nd: $MAPbI_3$ precursor solution was spin-coated on the quartz slide substrate to obtain the 2%Nd: $MAPbI_3$ thin film. The thin film is annealed at 120 °C for 10 min. Then a quartz slide was used to scratch the thin film into the powder which was then collected in a quartz tube for EPR measurement.

## X-ray diffraction (XRD)

The X-ray diffraction data were collected by X-ray diffractometer with Cu K$_\alpha$ radiation (Bruker D8 Advance A25, λ = 1.54060 Å). The scanning range was 10° to 80° (2θ). The data were processed to analyze all the samples by using the Rietveld method in the GSAS-II software for structure refinement.

## Transmission electron microscopy (TEM) and electron diffraction

High-resolution TEM and selected area electron diffraction (SAED) characterizations of the samples were completed on a JEOL JEM-2100 at an acceleration voltage of 200 kV.

## Inductive coupled plasma-mass spectrometry (ICP-MS)

In order to identify the element and determine the Nd: Pb in a molar ratio in the perovskite film, an Inductively Coupled Plasma Mass Spectrometer was used (Shimadzu ICPMS-2030 series ICP-MS instrument). The perovskite film prepared in the previous procedure was added to a plastic vial, and then 1 mL of concentrated $HNO_3$ was added to this vial to dissolve the perovskite film into the solution. Insert the vial into the centrifugal machine for 2 min and grab 200 μL liquid supernatant, and the predigest solution was made. Use a pipette to suck up the different volumes of predigest solution into the deionized water to make 50 mL sample solutions with different concentrations (100 ppm, 1000 ppm, 2000 ppm). The different concentrations of the 50 mL sample solution vials were inserted into the sample room for the ICP-MS test.

## X-ray photoelectron spectroscopy (XPS)

XPS was conducted in a PHI 5000 VersaProbe II system (Physical Electronics) attached to an Ar-atmosphere glovebox. The spectra were obtained using an Al Kα radiation (hυ = 1486.6 eV) beam (size = 100 μm*100 μm, power = 25 W), with Ar$^+$ and electron beam sample neutralization, in a fixed analyzer transmission mode. All the spectra have been calibrated against adventitious C 1s, C-C peak at 284.8 eV.

## Ultraviolet photoelectron spectroscopy (UPS)

The UPS measurement was performed using a Specs PHOIBOS 150 hemispherical energy analyzer and He I UV excitation (hυ = 21.22 eV). Samples were mounted in a glove box and transferred through an interconnected ultra-high-vacuum linear transfer system to the UPS system to avoid any exposure to air. The samples were grounded to the spectrometer through the Au substrate. The UPS spectra were measured using a pass energy of 2 eV at a resolution of 0.02 eV/step and a total integration time of 0.1 s/point. Spectra were charge referenced to the Fermi edge of the Au substrate centered at 0 eV.

## Continuous-wave electron paramagnetic resonance (CW-EPR)

The EPR data were recorded on a (Elexsys E500 CW-EPR) X-band Continuous wave (CW) electron paramagnetic resonance (EPR) spectrometer at the Center for Nanoscale Materials (CNM) at Argonne National Laboratory (ANL). A temperature controller (MercuryiTC by Oxford Instruments) was used for low-temperature measurements. The complete system was operated by Bruker Xenon software. The X-band frequency is 9.38 GHz, microwave power = 2 mW, and 100 kHz modulation amplitude is 10 G. The temperature for the EPR measurement was set to 4 K, 20 K, 100 K, 200 K, and 295 K. The simulation result is processed by MATLAB using the software package Easy Spin.

## Steady-state photoluminescence (ss-PL) and time-resolved photoluminescence (tr-PL)

ss-PL was performed on perovskite thin films on quartz from the quartz side with 400- nm laser from a diode laser or light source within a streak camera system (Hamamatsu C5680) at the Center for Nanoscale Materials (CNM) at Argonne National Laboratory. For cryogenic ss-PL and tr-Pl measurements, film samples are fixed to the front side of the copper cold finger using copper tape (0.5 cm away from the thermal couple on the cold finger). For PL measurement under a magnetic field, a square-shape SmCo permanent magnet (1 × 1 × 0.25 inch, Magnetshop) with its field normal to its square surface is attached using copper tape and thermal grease to the rear side of the cold finger behind the sample, so that the magnetic field is also normal to the sample surface. In this configuration, the magnetic field strength on the sample surface is measured to be about 1500 gauss by a Digital Tesla Meter (Tunkia TD8620). The system is then cooled down overnight to stabilize at about 5 K. The measurement under a magnetic field is then conducted at each set temperature (stabilized for at least 1 h) till reaching room temperature. Then the chamber is vented to remove the magnet. The system is vacuumed and cooled again to acquire the PL spectra on the same samples without a magnetic field. ss-PL from laser diode as the light source was obtained from 400 nm excitation wavelength, 1 nJ/cm$^2$ pump fluence, 35 fs pulse width and 2000 Hz repetition rate. For tr-PL experiments based on diode laser excitations, emission photons at center wavelengths were collected with a lens and directed to a grating spectrograph with 300 mm focal length as outfitted with a thermoelectrically cooled CCD and avalanche photodiode with time-correlated single-photon counting (TCSPC) electronics, where TimeHarp 260 software is used to record the tr-PL decay data with 0.2 ns per bin. In terms of ss-PL and tr-PL experiments with streak camera system, HPDTA software is used to process the PL images and decay dynamics, where ±10 nm of ssPL center wavelengths are taken as ranges from streak camera images to generate tr-PL decay dynamics. The average lifetime <τ> of PL is calculated by

$$<\tau> = \frac{a_1\tau_1{}^2 + a_2\tau_2{}^2}{a_1\tau_1 + a_2\tau_2} \quad (2)$$

The average lifetime <τ> is given by using the two-exponential decay expressed in Eq. 2, where $\tau_1$ and $\tau_2$ are the decay times, $a_1$ and $a_2$ are the pre-exponential factors.

## Scanning tunneling spectroscopy

For the measurement, 100 nm Au was coated on the quartz by thermal deposition method. After the plasma process, the pristine $MAPbI_3$ and 2%$Nd:MAPbI_3$ precursor were spin-coated on the Au-coated quartz substrate to form the thin film. STM experiment was performed on Createc ultra-high vacuum (UHV) low-temperature STM (LT-STM) at CNM/NST. After samples were loaded into the preparation chamber, it is heated to ~80 °C under UHV conditions for about 30 min to clear the adsorbed gas on the surface. Then samples were cooled down to 100 K and 5 K for the respective measurements. 400 nm excitation laser (~1 mW) at ~20 °C (from the surface of the sample) was used during the measurement. Tunneling spectroscopy (dI/dV) was performed using an external lock-in amplifier (SR830) at 1.01 kHz and 20 mV modulation with 30 ms integration time. STM tip was held and fixed above the sample using 1 V bias at 50 pA tunneling current during the dI/dV spectrums were collected.

## Magnetic properties measurement

The magnetic susceptibility and magnetization data were recorded on a Quantum Design MPMS XL at CNM at Argonne National Laboratory. By using a Superconducting Quantum Interference Device (SQUID) Magnetometer, the small changes in magnetic flux were detected with the intensity change of the external magnetic field. The temperature for the measurement was set to 5, 20, 50, 100, 120, 140, 170, 200, 250, 295 K.

## Data availability

The data that support the findings of this study are provided in the main text and the Supplementary Information. The original data are available from the corresponding author upon request.

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

## Acknowledgements

T.X. acknowledges support from the U.S. National Science Foundation (DMR 1806152). Work performed at Argonne National Laboratory's Center for Nanoscale Materials (CNM), US DOE Office of Science User Facilities, was supported by the U.S. DOE, Office of Basic Energy Science, under Contract No. DE-AC02-06CH11357. Z.Y. acknowledges the support from the Post Test Facility of Argonne National Laboratory, supported by the Department of Energy, Vehicle Technologies Office, under Contract No. DE-AC02-06CH11357. K.Z.L. and S.W.H. acknowledge the support of the U.S. Department of Energy, Office of Science, Office of Basic Energy Sciences, Materials Science and Engineering Division for the STM characterizations. T.X. also thanks the initial help from Dr. Peijun Guo at Yale University and the inspiring discussions with Dr. Qianfan Chen and Dr. Pierre Darancet at CNM, and Dr. Oleg Poluektov and Dr. Jens Niklas at Chemical Science and Engineering Division of Argonne National Laboratory.

## Author contributions

T.X. designed and supervised the research. X.X synthesized, fabricated, and characterized perovskite thin films and analyzed the data. Z.K.L. and S.W.H. conducted the STM study, analyzed the data, and provided insight into the mechanism. T.K. and J.G.C. conducted the UPS study and analyzed the data. C.H.F. conducted an EPR study. Y.L. conducted the TEM study. J.E.P. conducted the magnetization measurement. O.S.W. conducted an XRD study. M.L. and C.S. helped with synthesis. Z.Y. conducted an XPS and SEM study. B.T.D. and J.G. guided the PL study and analyzed the data. B.T.D. and R.D.S. provided insights into the mechanism. T.X., J.G., and X.X. wrote the manuscript. All authors discussed the results and contributed to manuscript preparation.

## Competing interests

The authors declare no competing interests.
