## [Peer Review File · Nature Communications]

REVIEWER COMMENTS

Reviewer #1 (Remarks to the Author):

In the study titled "Light-induced Kondo-like Exciton-Spin Interaction in Neodymium(II) Doped Hybrid Perovskite" the authors investigate the intriguing influence of neodymium doping on the photoluminescence (PL) response of MAPbI₃. While I find the paper generally interesting, I believe it falls short of delivering a comprehensive explanation for the observed phenomena.

The PL response suggests that carriers/excitons undergo a form of trapping, but the paper lacks a deeper elucidation of these phenomena. Notably, there is a lack of clarification on why excitons cease to be trapped by Nd²⁺ when aligned in the presence of a magnetic field. Furthermore, the paper does not establish with certainty whether the applied magnetic field aligns all Nd²⁺ magnetic moments. I propose that the impact of the magnetic field should be systematically examined to better understand its role.

Additionally, the use of circularly polarized light excitation could offer insights by selectively photoexciting excitons with specific spin orientations. This approach might enhance our understanding of the observed phenomena. It is essential to acknowledge that a comparable PL behaviour can be observed in the case of magnetic polaron formation (DOI: 10.1103/PhysRevB.83.081306). The paper should address whether this is the same effect, as introducing the new term "Kondo-like" exciton-spin interaction may not be justified if the effects are identical. If they differ, the paper should clearly delineate the distinctions between exciton-spin interaction in perovskites and what is known from diluted magnetic semiconductors.

In conclusion, while I appreciate the experimental observations presented in the work, the absence of a more comprehensive explanation renders this study, in my opinion, premature and falling short of the standards expected for publication in Nature Communications.

Reviewer #2 (Remarks to the Author):

The authors reported the study of kondo-like exciton-spin interaction in Nd doped MAPbI₃, which exhibit tunable carrier lifetime via Nd concentration or external magnetic field. Observing interaction of exciton with localized spins is interesting. However, a theoretical background and direct experimental evidences is missing to claim the kondo-like interaction, and the paper is not suitable for publication at nat commun at present stage.

1. The major evidence in the paper is the elongated PL lifetime in Nd doped MAPbI₃, by ~ 1 order. However, there are numerous mechanisms that could lead to elongated lifetime (Light Sci Appl 9, 192 (2020)). Thus, the discussion can not exclude other possibilities.
2. Only 2 samples with different Nd doping concentration is fabricated, the authors then use Nd²⁺/photon as a reference in discussing the lifetime. In total, 4 data points are collected for each temperature in Figure 2e, which is not enough for a solid discussion on the doping/illumination dependence. In my opinion, the authors had to validate the dependence by comparing it to measurements that changes light illumination only, and samples that changes Nd dopants.
3. dI/dV spectroscopy is used to compare Nd doped samples to the pristine one. However, the narrowing of bandgap can be artifacts at grain boundaries. There is no evidence that the probed dI/dV spectroscopy is related to Nd site due to the missing of surface features. According to Figure S4, the film structure is polycrystalline, what is the morphology of the crystal in high magnification?
4. What is the potential role of elongated PL lifetime in application by the claimed kondo-like effect is not discussed, making the paper lack deep insights. Could it be harness to improve the quantum yield in photoluminescence?

Responses to reviewers of manuscript NCOMMS-23-48438

Reviewer #1 (Remarks to the Author):

In the study titled "Light-induced Kondo-like Exciton-Spin Interaction in Neodymium(II) Doped Hybrid Perovskite" the authors investigate the intriguing influence of neodymium doping on the photoluminescence (PL) response of MAPbI₃. While I find the paper generally interesting, I believe it falls short of delivering a comprehensive explanation for the observed phenomena.

Response: We are thankful to the reviewer's efforts in evaluating our manuscript. Please see below point-by-point responses to your detailed comments.

1. The PL response suggests that carriers/excitons undergo a form of trapping, but the paper lacks a deeper elucidation of these phenomena. Notably, there is a lack of clarification on why excitons cease to be trapped by Nd²⁺ when aligned in the presence of a magnetic field. Furthermore, the paper does not establish with certainty whether the applied magnetic field aligns all Nd²⁺ magnetic moments. I propose that the impact of the magnetic field should be systematically examined to better understand its role.

Response: First, as inferred by time-resolved PL decay studies and the corresponding mechanistic picture (Figs. 2e-2g), excitons cease to be trapped when the magnetic moments of excitons' electrons/holes can no longer form antiferromagnetic exchange interaction with the net spin magnetic moments of Nd²⁺ 4f electrons. To elucidate if the applied magnetic field (1500 Gauss) aligned all Nd²⁺ magnetic moments, we further performed magnetization studies of both pristine and Nd-doped MAPbI₃ film samples, with the results shown in Supplementary Fig. 12. It can be seen that the applied magnetic field indeed was adequate in directing all Nd²⁺ magnetic pivots under 20 ppm and 2% doping concentrations.

Revisions made:

1. Highlighted revisions have been made on page 14 of the manuscript as "We further studied the PL under a fixed permanent magnetic field. First, we methodically studied the magnetization property of 2%Nd:MAPbI₃. The temperature-dependent hysteresis curve of 2%Nd:MAPbI₃ and pristine MAPbI₃ are collected in **Supplementary Fig. 12a**. It indicates that there is a positive magnetic moment observed, which can be attributed to the alignment of spin moments of Nd ions within the sample. With the decrease of temperature, the magnetization curves displayed nearly S-shaped behaviors, indicating stronger magnetic interactions. In contrast, the pristine MAPbI₃ shows diamagnetic behaviors showing a small negative response to the external magnetic field. Furthermore, we also studied the magnetic susceptibility as a function of temperature for the 2%Nd:MAPbI₃ powder, shown in **Supplementary Fig. 12b**. By using the Curie-Weiss (CW) law, the effective magnetic moment can be derived as $\mu_{\text{eff}} = 4.76 \mu_{\text{B}}$. This result is similar to the theoretical effective magnetic moment of Nd²⁺ which is $\mu_{\text{eff}} = 4.9046 \mu_{\text{B}}$. The result also indicates the bivalent state of Nd ion in the sample which corresponds to the XPS result."

2. Additionally, the use of circularly polarized light excitation could offer insights by selectively photoexciting excitons with specific spin orientations. This approach might enhance our understanding of the observed phenomena. It is essential to acknowledge that a comparable PL behaviour can be observed in the case of magnetic polaron formation (DOI: 10.1103/PhysRevB.83.081306). The paper should address whether this is the same effect, as introducing the new term "Kondo-like" exciton-spin interaction may not be justified if the effects are identical. If they differ, the paper should clearly delineate the distinctions between exciton-spin interaction in perovskites and what is known from diluted magnetic semiconductors.

Response: Previous literature has shown that exciton spin relaxation times in MAPbI₃ are rapid: reported decays using spin polarized transient absorption show a spin lifetime of <1.5 ps (J. Phys.

Chem. Lett. 2018, 9, 2595–2603) This is much faster than the time scale of processes responsible for light emission, which are nanosecond or much longer (tens of ns to μ s) in our system. Although measurements at low temperatures are expected to increase the spin lifetime, demonstrated examples suggest it is still less than 10 ps. (Nano Lett. 2015, 15, 3, 1553–1558) Because this is comparable to or shorter than the achievable temporal resolution in PL, spin relaxation dynamics will be washed out by the instrument response function. With such reasons, performing time-resolved PL using circularly polarized excitation will infer spin relaxation dynamics that are easily washed out by the intrinsic instrument response function. Therefore, we decided not to conduct such experiment.

Regarding the possibility of a magnetic polaron explanation, the reviewer is correct that magnetic polarons that were reported in doped conventional quantum-dot semiconductors, whereas the magnetic polarons are suppressed in the case of bulk semiconductor systems such as methylammonium lead iodide used in this work (Magnetic polarons, Introduction to the Physics of Diluted Magnetic Semiconductors, DR Yakovlev, W Ossau, 2010•Springer.). The physical observables in our system are distinct from the magnetic polarons in other dilute magnetic semiconductors, as the former requires optical excitation to generate photocarriers (excitons) to imitate the conduction electrons as occurring in classic metal-based Kondo systems, and arise from alignment of the localized dopants spins with the exciton spin. While the latter describes functional materials as formed by doping magnetic ions in conventional non-magnetic semiconductors (GaN, ZnO, etc.) to exhibit magneto-optical and magneto-electric properties that are not necessarily photoexcitation required. Magnetic polarons are characterized in particular by the Stokes shift that is transient redshift of photoluminescence, which is not observed in this case. Additionally, reported magnetic polarons result in faster photoluminescence lifetime (Physica Status Solidi b, 2002, 229, 473–476), contrary to the observed phenomenon in our work. Since the host material used in our work is novel semiconductor—organic-inorganic hybrid perovskite with well-defined optical bandgap, it is surely a type of novel diluted magnetic semiconductors.

Revisions made:

1. In order to better compare the PL behaviors as caused by magnetic polaron formation and exciton-spin interaction as observed in our work, we have made corresponding revisions as highlighted on pages 11 & 12 of the manuscript as “Although similar PL decay behaviors caused by magnetic polaron formation can be observed in semiconductors doped with magnetic ions, (38) they are still different from our finding as the former generates very short spin relaxation dynamics at the scale of picoseconds, and previous report has indicated a suppressed magnetic polaron in bulk semiconductor systems. On the other hand, as our material paradigm requires optical excitation to generate photocarriers (excitons) to imitate the conduction electrons as occurring in classic metal-based Kondo systems, and arise from the alignment of the localized dopants spins with the exciton spins, they also differ from the intrinsic magneto-optical and magneto-electric properties of diluted magnetic semiconductors.”.

In conclusion, while I appreciate the experimental observations presented in the work, the absence of a more comprehensive explanation renders this study, in my opinion, premature and falling short of the standards expected for publication in Nature Communications.

Response: We hope that the abovementioned responses have formed a more comprehensive explanation to address the scientific mechanisms and your concerns.

Reviewer #2 (Remarks to the Author):

The authors reported the study of kondo-like exciton-spin interaction in Nd doped MAPbI₃, which exhibit tunable carrier lifetime via Nd concentration or external magnetic field. Observing interaction of exciton with localized spins is interesting. However, a theoretical background and direct experimental evidences is missing to claim the kondo-like interaction, and the paper is not suitable for publication at nat commun at present stage.

Response: We want to thank the reviewer for the critical comments regarding our work. We have carefully gone through and responded to the concerned issues in a point-by-point manner as detailed below.

1. The major evidence in the paper is the elongated PL lifetime in Nd doped MAPbI₃, by ~ 1 order. However, there are numerous mechanisms that could lead to elongated lifetime (Light Sci Appl 9, 192 (2020)). Thus, the discussion can not exclude other possibilities.

Response: Indeed, there are multiple physical mechanisms that can lead to elongated PL lifetimes in perovskite materials. As detailed in the literature (Light Sci. Appl. 2020, 9, 192) mentioned by the reviewer, one can summarize several factors that affect photocarrier relaxation—1) Coulomb interactions, 2) initial distribution of photocarriers in electronic band structures, 3) defect-assisted relaxation, 4) phonon-assisted relaxation, 5) van der Waals structures, and 6) transition among different quasiparticles.

For Coulomb interaction, we consider it does not exactly apply to our scheme, where Coulomb screening mainly occurs when additional charge screening environments are composed, especially between the 2D semiconducting layer and dielectric environment, or between 2D layer and injected charged carriers; while the former case readily does not stand in our scenario, the latter one is also not applicable since we utilized identical photoexcitation conditions and is free from electrical bias, with our samples being isotropic—**3D perovskites**, contrastively different from the 2D layered material systems.

Regarding the initial distribution of photocarriers in electronic bands, since we measured the doped and undoped perovskite films under the same optical excitation and acquisition conditions, we thus consider the photocarrier density as generated in electronic bands of both samples

comparable at the same level. In addition, UPS spectra of both pristine and doped films nearly overlap in the range of Fermi level (B.E. = 0 eV) to Nd^{2+} 4f level (B.E. = 3.50 eV) (Fig. 1d), which indicates that density of states, in other words, electronic band structures are comparable across these two perovskite films. Based on these two folds of interpretation, we can deduce that the initial distribution of photocarriers in the electronic bands of both types of perovskite samples does not play a determining role in affecting the observed PL lifetimes.

For defect-assisted photocarrier relaxation, we consider it rather play a small role, as pristine and Nd-doped MAPbI_3 show comparable steady-state PL intensities and PL lifetimes at room temperature; plus, both types of perovskite films exhibit the same crystal structures at different temperatures (Fig. 1a & 1b, Supplementary Fig. 3a & 3b) as inferred by TEM/SAED studies and the corresponding diffraction analysis of perovskite samples. Therefore, it can be deduced that Nd doping does not phenomenally eliminate or create structural defects that could impact the photocarrier relaxation dynamics. While our work features bulk perovskite film, it is free from van der Waals structures and the corresponding interlayer exciton recombination scenarios.

We acknowledge that various influencing factors play a role in the relaxation process of photocarriers, including the relaxation process assisted by phonons. With this in mind, we examined the disparity in electron-phonon coupling between the pristine MAPbI_3 sample and the Nd-doped MAPbI_3 sample through an analysis of the temperature-dependent steady-state PL data. The broadening of emission observed in steady-state PL data has been extensively utilized to analyze the mechanism of electron-phonon coupling. To understand phonon-assisted photocarrier relaxation, we have analyzed the temperature-dependent emission broadening and extracting the full width at half-maximum (FWHM) of the PL spectra (Supplementary Fig. 11a). For hybrid lead halide perovskites, the Fröhlich interaction between charge carriers and LO phonons provides predominantly linewidth broadening (Nat. Commun. 2016, 7, 11755). Then we calculated the temperature-dependent energy (Γ_{LO}) caused by the LO phonon scattering with the charge-carrier phonon of pristine and Nd-doped MAPbI_3 films as extracted from the spectral widths of their temperature-dependent steady-state PL spectra and fitting results given by the Bose-Einstein distribution function. Results indicate a smaller Γ_{LO} of Nd-doped MAPbI_3 films as compared to pristine counterparts (Supplementary Fig. 11b), thus indicating a suppressed carrier-phonon coupling that should contribute to retarded photocarrier relaxation. But given the magnetic-field manipulation effects (not affecting the optically relevant phonons), we can deduce that the underlying governing effects are still the spin-dependent Kondo-like interaction that slows the exciton recombination process.

In terms of van der Waals structures that affect photocarrier relaxation, they only exist in 2D layered materials, much similar to the case of Coulomb interaction.

As for transition among different quasiparticles (exciton, trion, free carriers) that influences photocarrier relaxation, it requires gate voltage to generate free carriers and does not apply to our work, where steady-state PL spectroscopy and time-resolved PL decays are contact-free optical studies. On the other hand, trion typically exists as interlayer excitons in multilayer

heterostructures (e.g. MoSe₂/WSe₂ in the mentioned literature), a scheme that is also not applicable to our work.

Revisions made:

1. To better reflect a more complete discussion of the mechanistic pictures that contribute to elongated PL lifetimes, we have made revisions as highlighted on pages 12-13 as

“It is noteworthy that other factors could lead to elongated PL carrier lifetimes, such as Coulomb interactions, initial distribution of photocarriers in electronic bands, defect-assisted relaxation, van der Waals structures, and transition among different quasiparticles. (39) But we can mostly exclude these factors because 1) perovskite materials applied in this study are three-dimensional structures with and without atomically dispersed Nd²⁺ dopant, and therefore do not introduce multilayered heterostructures, external dielectric environment or charge carriers; 2) identical photoexcitation and acquisition conditions are utilized, plus the Nd energy levels that mostly affect perovskite intragap and VB regions, both pristine and Nd-doped perovskite films exhibit comparable electronic band structures; 3) both samples show comparable ss-PL intensities and tr-PL decay dynamics initially before temperature reduction, and temperature-dependent TEM/SAED images and the derived diffraction patterns (**Supplementary Fig. 3b**) indicate comparable crystal structures, thus signifying comparable densities of structural defects; 4) at cryogenic temperatures, the form of photocarriers remains excitons in the absence of electrical bias or multilayer heterostructures. On the other hand, to comprehend the relaxation of phonon-assisted photocarriers, we examined the temperature-dependent broadening of emissions and extracted the full width at half-maximum (FWHM) from the photoluminescence spectra. Through the analysis of the temperature-dependent emission broadening and extracting the full width at half-maximum (FWHM) of the PL spectra (**Supplementary Fig. 11a**), one can see that Nd-doped MAPbI₃ film indeed exhibits a smaller change of the temperature-dependent PL linewidth compare with the pristine MAPbI₃ film. For hybrid lead halide perovskite materials, charge-carrier-phonon interaction is dominated by Fröhlich interaction between charge carriers and LO phonons, (40) the equation is expressed as:

$$\Gamma_{LO} = \gamma_{LO} \cdot \frac{1}{[e^{-E_{LO}/k_B T} - 1]} \quad (1)$$

Here in **Equation 1**, Γ_{LO} is results from LO phonon scattering, γ_{LO} is the corresponding charge-carrier-phonon coupling strength, and E_{LO} is the Energy representative of the frequency for the weakly dispersive longitudinal optical (LO) phonon branch. The parameters, γ_{LO} and E_{LO} can be derived by fitting the temperature-dependent half-maximum (FWHM) of the PL spectra. Therefore, the longitudinal optical phonon energy (Γ_{LO}) of Nd-doped MAPbI₃ as compared to pristine MAPbI₃ at different temperatures (**Supplementary Fig. 11b**) can be exported, thereby substantiating the suppressed carrier-phonon coupling mentioned in the previous context.”.

Supplementary Fig. 3. Temperature-dependent SAED patterns. **b**, Comparison of the SAED patterns between pristine MAPbI₃ and 2%Nd:MAPbI₃ at different temperatures from 20K to room temperature, and corresponding SAED analysis by using Digital Micrograph software. Based on the SAED patterns analyzed in Digital Micrograph software, the crystal structure of pristine MAPbI₃ and 2% Nd:MAPbI₃ remain similar at the same temperature. The characteristic peak of the orthorhombic phase at (221) crystal plane was found at 100K. However, at 200 K, this crystal planes disappear while other planes characteristic to the tetragonal phase are present including (004) and (220) planes. We thus conclude that the phase transition from orthorhombic to tetragonal is around 100~160 K, in agreement with reports in literature.

2. We have also added Supplementary Fig. 11a to compare the temperature-dependent FWHM variation of steady-state PL peaks of both pristine and Nd-doped MAPbI₃ films, and also the Supplementary Fig. 11b, the temperature-dependent energy (Γ_{LO}) of both pristine and Nd-doped MAPbI₃ films to understand the electron-phonon coupling strength.

Supplementary Fig. 11. FWHM analysis of the temperature-dependent emission broadening spectra. a, FWHM of PL peak vs temperature for pristine MAPbI₃ and 2%Nd:MAPbI₃ films. b, LO phonon (Fröhlich) scattering vs temperature for pristine MAPbI₃ and 2%Nd:MAPbI₃ films.

2. Only 2 samples with different Nd doping concentration is fabricated, the authors then use Nd²⁺/photon as a reference in discussing the lifetime. In total, 4 data points are collected for each temperature in Figure 2e, which is not enough for a solid discussion on the doping/illumination dependence. In my opinion, the authors had to validate the dependence by comparing it to measurements that changes light illumination only, and samples that changes Nd dopants.

Response: As per the request of reviewer, we have further supplemented PL lifetimes of MAPbI₃ films based on a low [Nd²⁺] doped 20ppb-Nd:MAPbI₃ (i.e. molar ratio of Nd:Pb =20:10⁹) concentration obtained at two excitation fluences (1760 uW/cm² and 220 uW/cm²) in Fig. 2c, corresponding to 10⁻³ and 10⁻² [Nd²⁺]_{area}/[photon], in order to enhance the analytical legitimacy of doping/illumination dependence.

Revisions made:

1. To fully validate the dependence of PL lifetime with respect to Nd²⁺ concentration/photon fluence, we have made highlighted revision on page 13 as “Fig. 2c shows the temperature impact on the $\langle \tau \rangle$ across the [Nd²⁺]_{area}/[photon] range of 10⁻³~10⁴. The monotonic increment of $\langle \tau \rangle$ at greater [Nd²⁺]_{area}/[photon] ratio below 100 K is clearly evident”.

2. Fig. 2c is revised with the addition of 16 data points for 10⁻³ and 10⁻² [Nd²⁺]_{area}/[photon] ratio.

3. *dI/dV* spectroscopy is used to compare Nd doped samples to the pristine one. However, the narrowing of bandgap can be artifacts at grain boundaries. There is no evidence that the probed *dI/dV* spectroscopy is related to Nd site due to the missing of surface features. According to Figure S4, the film structure is polycrystalline, what is the morphology of the crystal in high magnification?

Response: The characteristic step-like features observed in Fig. 3e are only observed in Nd-doped MAPbI₃ samples, which is related to the inelastic electron tunneling (IET) process. In stark contrast, this step-like feature was never observed on pristine MAPbI₃ film. The corresponding statistic data are presented in Supplementary Figs. 15-17.

Revisions made:

1. On page 18, “More complete tunneling spectroscopic data collected at random sample areas reveal both the presence and absence of step-like features in dI/dV and d^2I/dV^2 curves of Nd-doped MAPbI₃ film (**Supplementary Fig. 15**). **Supplementary Fig. 16** exhibits another local domain with more uniform distribution of Nd²⁺ as evidenced by the periodical occurrence of the step-like features. In stark contrast, this step-like feature was never observed from pristine MAPbI₃ film that only exhibits the typical bandgap characteristics of MAPbI₃ (**Supplementary Fig. 17**), suggesting that the IET process only occurs in Nd-doped MAPbI₃ film.”

2. Supplementary Figs. 15-17 have been added into the Supporting Information.

Supplementary Fig. 15a. Tunneling spectroscopy of Nd²⁺-doped MAPbI₃ perovskite with local sampling. An STM image of a Nd²⁺ doped sample surface acquired under 400 nm illumination ($V_t = 1$ V, $I_t = 50$ pA).

Supplementary Fig. 15a shows an STM image of a Nd²⁺-doped MAPbI₃ perovskite sample. Here, the distances between the grid points are 1 nm x 1 nm. The tunneling spectroscopy data were recorded under 400 nm illumination. The green number locations give a semiconducting gap (~ 1.5 V), while the red number locations exhibit a smaller gap-like feature and show symmetric peaks in d^2I/dV^2 data.

Supplementary Fig. 15b. I-V (left), dI/dV (middle), and d^2I/dV^2 (right) curves of Nd^{2+} -doped MAPbI_3 perovskite with local area sampling. Here, the spectroscopy numbers at the left correspond to the numbers shown in Supplementary Fig. 15a. Note that at the location ‘8’, only a small gap-like feature is observed (indicated with arrows).

Supplementary Fig. 16a. Tunneling spectroscopy of Nd^{2+} -doped MAPbI_3 perovskite with spatially periodic sampling. An STM image of a Nd^{2+} -doped sample surface acquired under 400 nm illumination ($V_t = 1$ V, $I_t = 50$ pA).

Supplementary Fig. 16a shows an STM image of a Nd^{2+} -doped MAPbI_3 perovskite sample. Here, the distances between the grid points are 2 nm x 2 nm. The tunneling spectroscopy data were recorded under 400 nm illumination, and all the spectroscopy recorded at this location exhibited a gap-like feature (see the following spectroscopic data).

Supplementary Fig. 16b. I-V (left), dI/dV (middle), and d^2I/dV^2 (right) curves of Nd^{2+} -doped MAPbI_3 perovskite with spatially periodic sampling. Here, the spectroscopy numbers at the left correspond to the numbers shown in Supplementary Fig. 16a.

Supplementary Fig. 17a. Tunneling spectroscopy of pristine MAPbI₃ perovskite with spatially periodic sampling. An STM image of an undoped sample surface acquired under 400 nm illumination ($V_t = 1$ V, $I_t = 50$ pA).

Supplementary Fig. 17a shows an STM image of an undoped MAPbI₃ perovskite sample. Such images were acquired at the different locations on the undoped samples, and I-V, dI/dV, and d²I/dV² tunneling spectroscopy data were simultaneously recorded on 8 x 8 grid points at each surface area. Here, the distances between the grid points are 2 nm x 2 nm. The tunneling spectroscopy data were recorded under 400 nm illumination, and the undoped samples usually provide a bandgap and exhibit a semiconducting behavior. An example spectroscopy data sequence measured at this sample location is provided below.

Supplementary Fig. 17b. I-V (left), dI/dV (middle), and d²I/dV² (right) curves of pristine MAPbI₃ perovskite with spatially periodic sampling.

4. *What is the potential role of elongated PL lifetime in application by the claimed kondo-like effect is not discussed, making the paper lack deep insights. Could it be harness to improve the quantum yield in photoluminescence?*

Response: We are grateful to the insightful request raised by the reviewer. The potential role of elongated PL lifetime, as achieved in this work by the Kondo-like effect between perovskite excitons and the spins in Nd^{2+} impurity, is to construct a **long** and **manipulable** spin relaxation process that could enable spintronics and quantum computing with greater performances, as we have originally pointed out in the Introduction section of the manuscript as “As such, Kondo-like interactions between the impurity spin and the light-induced exciton containing a pair of spin-entangled electron-hole can render a potential optic platform for spintronics and spin-based quantum computing”. Optically, the much enhanced PL emission intensities at low temperatures with Nd^{2+} doping, also warrant a practical approach to enhance the PL quantum yield of bulk perovskite films, in contrast with the conventionally used nanocrystal thin films for large emission efficiency.

Revisions made:

1. As per the reviewer’s suggestion, we have added highlighted revisions to enhance the discussion of the potential role of the elongated PL lifetimes on page 14 “As such, the demonstrated elongated PL lifetime as controllable upon Nd^{2+} doping concentration and magnetic field, clearly indicate a long but manipulable spin relaxation process that are potentially useable in high-performance spintronics and quantum computing applications, where achieving long coherence time of electron spins is critical for quantum manipulation. (42)”.

2. We also address the phenomenon of notably enhanced PL emission intensities after Nd^{2+} doping on pages 10 & 11 as “Such enhanced PL intensities strongly suggest a potential application of Nd^{2+} doping in enhancing the PL intensities of bulk perovskite films, in contrast with the conventionally adopted quantum-dot perovskite films for their large emission efficiencies”.

REVIEWER COMMENTS

Reviewer #1 (Remarks to the Author):

The authors have made improvements to the paper following the revision process, yet in my assessment, the experimental evidence for the claimed Kondo-like interaction remains insufficiently comprehensive.

One notable gap is the lack of exploration into how the application of a magnetic field affects the photoluminescence (PL) spectrum. While the dynamic analysis has been emphasized, the absence of data on how the PL spectrum changes when the magnetic field is toggled is conspicuous. Given that the elongation of PL decay times supposedly correlates with the interaction with Nd spin, it is imperative to demonstrate whether the PL spectrum alterations occur when the magnetic field is turned on or off. This would help discern whether the increase in PL intensity and carrier lifetime is indeed unrelated to enhanced disorder subsequent to Nd incorporation, as the disorder can paradoxically enhance light emission efficiency in certain contexts, such as in InGaN [Appl. Phys. Lett. 74, 1460–1462 (1999), Science 281, 956–961 (1998)].

Furthermore, it is crucial for the authors to elaborate on how the magnetic field was applied. The vague description provided, stating that the magnet was "positioned behind the samples," raises doubts. For instance, if a cold finger cryostat was employed for the investigation, it could result in weak thermal contact and unreliable determination of sample temperature, potentially explaining the observed shortening of the PL decay time. Unfortunately, such pertinent details are absent from both the manuscript and supplementary information.

Regrettably, the article appears somewhat carelessly prepared. For instance, the streak camera data in the supplementary information consists of mere "screen-shots" from the software controlling the streak camera, and other figures suffer from readability issues.

In summary, the conclusions drawn in the manuscript lack robust support from the experimental data. The purported influence of magnetic properties on optical properties remains unsubstantiated. Consequently, I cannot recommend this paper for publication in Nature Communications.

Reviewer #2 (Remarks to the Author):

I am satisfied with the revision. The revised manuscript can be published in the journal now.

Responses to reviewers of manuscript NCOMMS-23-48438A

Reviewer 1: *The authors have made improvements to the paper following the revision process, yet in my assessment, the experimental evidence for the claimed Kondo-like interaction remains insufficiently comprehensive.*

Our response: We hope the additional revision adequately answered the reviewer's questions.

Reviewer 1: *One notable gap is the lack of exploration into how the application of a magnetic field affects the photoluminescence (PL) spectrum. While the dynamic analysis has been emphasized, the absence of data on how the PL spectrum changes when the magnetic field is toggled is conspicuous. Given that the elongation of PL decay times supposedly correlates with the interaction with Nd spin, it is imperative to demonstrate whether the PL spectrum alterations occur when the magnetic field is turned on or off. This would help discern whether the increase in PL intensity and carrier lifetime is indeed unrelated to enhanced disorder subsequent to Nd incorporation, as the disorder can paradoxically enhance light emission efficiency in certain contexts, such as in InGaN [Appl. Phys. Lett. 74, 1460–1462 (1999), Science281,956-961(1998)].*

Our response: We have made the following revisions:

On page 14 of the manuscript, highlighted revisions are made as “To verify the magnetic field effect on the PL spectra of the Nd²⁺ doped sample, we repeated the measurement on the same 2%Nd:MAPbI₃ sample. Comparison of the ss-PL spectra of this 2%Nd:MAPbI₃ sample with and without magnetic field illustrates the markedly attenuated PL intensities when the magnetic field is present (**Supplementary Fig. 13**). **Supplementary Fig. 14** verified the long PL lifetime in this 2%Nd:MAPbI₃ sample in absence of magnetic field at low-temperature range (<120K), similar to the 2%Nd:MAPbI₃ sample (no magnetic field) in **Fig. 2b**. However, under magnetic field, the long PL lifetime of this 2%Nd:MAPbI₃ sample in the low-temperature range (<120K) vanished, and becomes similar to that of the pristine MAPbI₃ sample (without magnetic field) as shown in **Fig. 2b**. On the other hand, we can rule out the possibility that Nd doping leads to lattice disorder and the potential increments in PL intensity and carrier lifetime, despite that structural distortion can in fact enhance light emission efficiencies in a certain inorganic context such as InGaN multilayer structure (42, 43). The magnetic field should not be able to control the regarded lattice disorder and annihilate/generate structural defects that affect the ss-PL intensity and carrier lifetime as observed. Therefore, the exciton-spin interaction as valved by the magnetic field and Nd impurity spins should be the responsible mechanism for the abovementioned light-induced observations at cryogenic conditions”.

Regarding the disordering-induced PL lifetime elongation, we have addressed this question in our

previous response letter. We consider structural defect-induced photocarrier relaxation plays a small role in our system, as both pristine and Nd-doped MAPbI₃ exhibit comparable steady-state PL intensities and PL lifetimes at room temperature. Moreover, both types of perovskite films demonstrate identical crystal structures at various temperatures, as evidenced by TEM/SAED studies and corresponding diffraction analyses (see Fig. 1a & 1b, Supplementary Fig. 3a & 3b). Hence, it can be inferred that Nd doping does not significantly introduce or eliminate structural defects that could impact photocarrier relaxation dynamics. The two references mentioned by Reviewer 1 focus on the inter-layer strain effect on the PL lifetime of epitaxial LED quantum well layers. In contrast, our MAPbI₃ is a 3D continuous structure without interface of the layers. On page 12, we stated that “1) perovskite materials applied in this study are three-dimensional structures with and without atomically dispersed Nd²⁺ dopant, and therefore do not introduce multilayered heterostructures, external dielectric environment, interlayer strain, or charge carriers;”

Reviewer 1: *Furthermore, it is crucial for the authors to elaborate on how the magnetic field was applied. The vague description provided, stating that the magnet was "positioned behind the samples," raises doubts. For instance, if a cold finger cryostat was employed for the investigation, it could result in weak thermal contact and unreliable determination of sample temperature, potentially explaining the observed shortening of the PL decay time. Unfortunately, such pertinent details are absent from both the manuscript and supplementary information.*

Our Response: We appreciate the reviewer being prudent and so are we. In order to better describe the experiment configuration. We have added the detailed experimental descriptions regarding how PL measurement under magnetic field was conducted as highlighted on page 14 as “See Materials and Methods section for the detailed descriptions of the magnetic field effect PL measurement procedures”, and page 24 under the section of “Materials and Methods” as: “For cryogenic ss-PL and tr-PL measurements, film samples are fixed to the front side of the copper cold finger using copper tape (0.5cm away from the thermal couple on the cold finger). For PL measurement under magnetic field, a square SmCo permanent magnet (1×1×0.25 inch, Magnetshop) with its field normal to its square surface is attached using copper tape and thermal grease to the rear side of the cold finger right behind the samples, so that the magnetic field is also normal to the samples’ surface. In this configuration, the magnetic field strength on the sample surface is measured to be about 1500 gauss by a Digital Tesla Meter (Tunkia TD8620). The system is then cooled down overnight to stabilize at about 5K. The measurement under magnetic field is then conducted at each set temperature (stabilized for at least 1 hour) till reaching room temperature. Then the chamber is vented to remove the magnet. The system is vacuumed and cooled again to acquire the PL spectra on the same samples without magnetic field.”

First of all, for review purpose only, we conducted a dynamic thermal analysis (Solidworks thermal analysis) per the experimental setup summarized in **Table R1** and **Figure R1**. We simulated an extreme case scenario that the temperature of LH Cryogen end is 5K, and it will take only 1500 seconds for the thermocouple, quartz sample, and magnet to reach 5K from room temperature, as shown in **Figure R2**.

Table R1. Simulated thermal properties of copper cold finger, SmCo magnet, and quartz substrate used in the cryogenic experimental setup of ss-PL/tr-PL measurements.

Material	Thermal conductivity (W/m·K)	Specific heat (J/kg·K)
Cu cold finger	390	390
SmCo magnet	10	200
Quartz sample (SiO ₂)	1.4	740

Figure R1. Schematic diagram of the cooling setup.

Figure R2. Simulated chronic cooling profile. The red curve is the cooling profile for Cu finger only; the blue finger is the cooling profile for all samples and the magnet attached.

However, in the real experiment condition, the ramp rate is controlled as 1K/min during the cooling down process via a heating element in the system. It takes 290 minutes too cooled down from 295K to 5K. **Figure R3** is the control panel of the Lake Shore 306 temperature controller, the heater output is changeable to control and stabilize the temperature. We always cool down the system overnight (15 hours) before the measurement starts, which is sufficiently long enough. Besides, during the temperature ramp-up process, we wait at least additional 60 min after the system reaches each desired temperature shown in the control panel.

Figure R3. The control panel of temperature controller.

Reviewer 1: *Regrettably, the article appears somewhat carelessly prepared. For instance, the streak camera data in the supplementary information consists of mere "screen-shots" from the software controlling the streak camera, and other figures suffer from readability issues.*

Our response: We admit that the scale bar is not clear enough to show the detailed data. The software does not allow us to directly extract the dataset and figures while only allowing us to analyze the data using the software. We have improved the visibility of the data by labeling the X and Y axes.

Reviewer 1: *In summary, the conclusions drawn in the manuscript lack robust support from the experimental data. The purported influence of magnetic properties on optical properties remains unsubstantiated. Consequently, I cannot recommend this paper for publication in Nature Communications.*

Our response: Scientific reasoning and formal logic have been extensively used in this manuscript to exclude other possibilities while our experimental evidence strongly supports the Kondo-like interaction. Therefore, we are confident that this is a Kondo-like spin-related phenomenon. We hope the revised manuscript gains the reviewer's satisfaction.

Reviewer 2: *I am satisfied with the revision. The revised manuscript can be published in the journal now.*

Our response: We are grateful to Reviewer 2's positive comment and satisfaction toward our revisions to the manuscript.

REVIEWERS' COMMENTS

Reviewer #1 (Remarks to the Author):

The authors have addressed my question, and I find their interpretation of the reported results convincing. However, I still have some reservations regarding the visual quality of the data presentation. Nonetheless, the findings are intriguing and merit publication.